# Four layer multi-omics reveals molecular responses to aneuploidy in *Leishmania*

**Bart Cuypers**[1,2], **Pieter Meysman**[2], **Ionas Erb**[3], **Wout Bittremieux**[4], **Dirk Valkenborg**[5,6], **Geert Baggerman**[6,7], **Inge Mertens**[6,7], **Shyam Sundar**[8], **Basudha Khanal**[9], **Cedric Notredame**[3], **Jean-Claude Dujardin**[1,10]*, **Malgorzata A. Domagalska**[1☯]*, **Kris Laukens**[2☯]*

1 Molecular Parasitology Unit, Department of Biomedical Sciences, Institute of Tropical Medicine, Antwerp, Belgium, 2 Adrem Data Lab, Department of Computer Science, University of Antwerp, Antwerp, Belgium, 3 Centre for Genomic Regulation (CRG), The Barcelona Centre of Science and Technology, Barcelona, Spain, 4 Skaggs School of Pharmacy and Pharmaceutical Sciences, University of California San Diego, La Jolla, California, United States of America, 5 Center for Statistics - University of Hasselt, Hasselt, Belgium, 6 Center for Proteomics, University of Antwerp, Antwerp, Belgium, 7 VITO, Mol, Belgium, 8 Institute of Medical Sciences, Banaras Hindu University, Varanasi, India, 9 BP Koirala Institute of Health Sciences, Dharan, Nepal, 10 Department of Biomedical Sciences, University of Antwerp, Antwerp, Belgium

☯ These authors contributed equally to this work.
* jcdujardin@itg.be (J-CD); mdomagalska@itg.be (MD); kris.laukens@uantwerpen.be (KL)

**Data Availability Statement:** The raw genomic and transcriptomic data in this study is available at the NCBI Sequence Read Archive (SRA) under the accession code: PRJNA762444. Raw proteomic

## Abstract

Aneuploidy causes system-wide disruptions in the stochiometric balances of transcripts, proteins, and metabolites, often resulting in detrimental effects for the organism. The protozoan parasite *Leishmania* has an unusually high tolerance for aneuploidy, but the molecular and functional consequences for the pathogen remain poorly understood. Here, we addressed this question *in vitro* and present the first integrated analysis of the genome, transcriptome, proteome, and metabolome of highly aneuploid *Leishmania donovani* strains. Our analyses unambiguously establish that aneuploidy in *Leishmania* proportionally impacts the average transcript- and protein abundance levels of affected chromosomes, ultimately correlating with the degree of metabolic differences between closely related aneuploid strains. This proportionality was present in both proliferative and non-proliferative *in vitro* promastigotes. However, as in other Eukaryotes, we observed attenuation of dosage effects for protein complex subunits and in addition, non-cytoplasmic proteins. Differentially expressed transcripts and proteins between aneuploid *Leishmania* strains also originated from non-aneuploid chromosomes. At protein level, these were enriched for proteins involved in protein metabolism, such as chaperones and chaperonins, peptidases, and heat-shock proteins. In conclusion, our results further support the view that aneuploidy in *Leishmania* can be adaptive. Additionally, we believe that the high karyotype diversity *in vitro* and absence of classical transcriptional regulation make *Leishmania* an attractive model to study processes of protein homeostasis in the context of aneuploidy and beyond.

data has been submitted to the PRIDE database under reference PXD028521. Raw metabolomic data was submitted to the MetaboLights database under accession code MTBLS1612 [67].

**Funding:** This work was supported by the Research Foundation Flanders (FWO) with post-doc fellowship 11O1614N to B.C. This work was also supported by the Interuniversity Attraction Poles Program of Belgian Science Policy (P7/41 to JC.D.), the InBev Baillet-Latour foundation (to JC. D.) and the Department of Economy, Science and Innovation in Flanders (ITM-SOFIB, SINGLE to JC. D.). The funders had no role in study design, data collection and analysis, decision to publish, or preparation of the manuscript.

**Competing interests:** The authors have declared that no competing interests exist.

## Author summary

*Leishmania* are protozoan parasites causing severe and stigmatizing diseases in humans and animals, worldwide. The parasites show several unique molecular features, one of them being an unusually high tolerance to aneuploidy, i.e. the presence of an abnormal number of chromosomes in the cell. While this is generally deleterious for higher Eukaryotes, this seems to be advantageous for *Leishmania* in certain conditions. However, the molecular and functional consequences of aneuploidy for the pathogen remained poorly understood. Here we studied the four main molecular layers of Eukaryotic cells: the genome, transcriptome, proteome, and metabolome. We present their first integrated analysis in aneuploid *Leishmania*. Our analyses show a strong impact of chromosome copy number on transcript- and protein abundance levels, ultimately correlating with the degree of observed metabolic differences. However, some specific proteins (subunits of protein-complexes and non-cytoplasmic proteins) encoded by aneuploid chromosomes showed reduced dosage effects. Reciprocally, we also found that some differential transcripts and proteins (chaperones, chaperonins and heat shock proteins) of aneuploid strains originated from non-aneuploid chromosomes. Our study provides new insights into mechanisms of molecular adaptation and regulation in *Leishmania* and the role of aneuploidy in this process.

## Introduction

Aneuploidy is the presence of an abnormal number of chromosomes in a cell. All genes encoded by aneuploid chromosomes undergo a shift in their copy number or "gene dosage". This large number of unbalanced dosage changes typically leads to broad-scale disruptions in the stoichiometric balances of transcripts, proteins, and metabolites, making aneuploidy most widely known for its detrimental effects. Indeed, aneuploidy can cause oxidative stress, misfolded protein stress (proteotoxicity), and metabolic stress at the cellular level [1]. In multicellular organisms, where cells are closely fine-tuned to work together, this often leads to growth defects or lethality [2,3].

Contrastingly, aneuploidy can also result in fitness gains. In fungi, specific karyotypes confer resistance against drugs, toxicants, or increase survival under nutrient limiting conditions [4–7]. Likewise, in cancer cells, aneuploidy can be tumor growth-promoting [8]. By altering the copy number of many genes at once, aneuploidy enables cells to explore a wide fitness landscape [9]. Particularly when facing very novel or competitive conditions, aneuploidy might allow the rapid selection of a beneficial phenotype from a pool of many divergent phenotypes [10]. Fully understanding this complex trade-off between aneuploidy fitness gains and losses requires a systems biological comprehension of its molecular impact.

In this context, a new and unique Eukaryote model is emerging, *Leishmania*. This protozoan parasite (Euglenozoa, Kinetoplastida) that is transmitted by sand flies and infects humans and various vertebrate animals shows a remarkably high tolerance for aneuploidy. A genomic analysis of 204 *in vitro* cultured *Leishmania donovani* strains uncovered up to 22 aneuploid chromosomes (out of 36) within a single strain [11], while *in vivo*, aneuploidy is less pronounced [12]. All 36 chromosomes have the capacity to become aneuploid and lead to viable parasites [13]. Strikingly, clonal populations of the parasite do not have uniform karyotypes but display a mosaic of them [14]. Single-cell sequencing of two clonal promastigote populations of *L. donovani* identified respectively 208 different karyotypes in 1516 sequenced cells (that were generated in approximately 126 generations) and 117 karyotypes in 2378 cells (56

generations). This high degree of genomic instability and mosaicism could explain the parasite's ability to rapidly select more advantageous karyotypes upon encountering novel *in vivo* or *in vitro* environments [12].

*Leishmania* also features an unusual gene expression system that is markedly different from most other Eukaryotes. Transcription of protein-coding genes cannot be controlled individually as these lack separate RNA polymerase II promoters. Instead, they are transcribed constitutively in long polycistronic units that span 10's to 100's of functionally unrelated genes [15]. The abundance of individual mRNAs and proteins is still regulated, but essentially post-transcriptionally [12,16]. Thus, the most straightforward evolutionary path towards altered gene expression might be altered gene dosage, which could explain the high incidence of aneuploidy in this organism [12]. While aneuploidy in *Leishmania* is known to directly affect the transcription of encoded transcripts, its molecular impact further downstream remains poorly understood [12,17].

Here we present the first study that generated and integrated genome, transcriptome, proteome, and metabolome data of highly aneuploid *L. donovani* strains, *in vitro*. This multi-omics analysis unambiguously demonstrates that aneuploidy in *Leishmania* globally and drastically impacts the transcriptome and proteome of affected chromosomes, ultimately correlating with the degree of metabolic differences between strains. This impact is not restricted to a single life stage as we observed it *in vitro* in both the parasite's proliferative and non-proliferative/infectious promastigote life stages. Interestingly, not all proteins responded equally to chromosome dosage changes; protein complex subunits and non-cytoplasmic proteins responded less or even not at all to aneuploidy. In parallel, the aneuploid *Leishmania* strains here studied also feature many differentially expressed proteins from non-aneuploid chromosomes. These were enriched for aneuploidy-related protein classes, including metabolism-related chaperones and chaperonins, peptidases, and heat-shock proteins. Through this integrated four 'omic approach, we provide a systems-wide view of the molecular changes associated with aneuploidy in *Leishmania*.

## Methods

### Parasites

The experiments in this study were performed on six cloned strains of *Leishmania donovani*: BPK275/0 cl18 (BPK275), BPK173/0 cl3 (BPK173), BPK282 cl4 (BPK282), BPK288/0 cl7 (BPK288), BPK178/0 cl3 (BPK178), BHU575/0 cl2 (BHU575) (S1A Table). A strain is a parasite population derived by serial passage *in vitro* from a primary isolate (from patient) with some degree of characterization, in our case bulk genome sequencing (definition modified from [18]). These strains were cloned to ensure genomic homogeneity. Parasites were cultured in HOMEM (Gibco) supplemented with 20% heat-inactivated fetal bovine serum (Biochrom), three days for replicative forms (LOG growth phase, 4–6 replicates) and six days for non-replicative forms (STAT, 4 replicates). Parasites from all replicates were washed twice by centrifugation at 18,000 x g at 0 ˚C, resuspension in ice-cold PBS buffer (PBS; pH 7.4 –Invitrogen), and pelleted at 18,000 x g at 0 ˚C. DNA, RNA, protein, and metabolite extractions were carried out in parallel (see further).

### DNA/RNA extraction, library preparation, and sequencing

DNA and RNA were extracted from the parasite pellets with the AllPrep DNA/RNA Mini Kit (Qiagen). The RNA integrity was then verified with the RNA 6000 Nano Kit using a Bioanalyzer 2100 (Agilent). DNA was quantified with the Qubit dsDNA BR assay (Thermo Fisher Scientific) and RNA with the Qubit RNA HS Assay.

Library preparation and sequencing of the DNA samples were performed at the Beijing Genomics Institute (BGI). Libraries were prepared with the TruSeq Nano DNA HT sample prep kit (Illumina) and 2 x 151 bp sequenced on the Illumina HiSeq 4000 platform. RNA sequencing libraries were prepared using the Spliced leader sequencing (SL-Seq) method as described in Cuypers et al. (2017) [16]. This protocol makes use of the presence of the affixed 39 nucleotide sequence spliced-leader (SL) that is present at the 5' end of all functional *Leishmania* mRNAs. RNAs containing a SL are selectively amplified with the protocol, and adapters required for Illumina sequencing are ligated. The SL-Seq libraries were 1 X 50 bp sequenced with the HiSeq 1500 platform of the Center of Medical Genetics Antwerp (Belgium).

## Genome and transcriptome data analysis

The quality of the raw sequencing data was verified with FASTQC v0.11.4 [19]. Reads were quality trimmed using Trimmomatic v035 [20] with the settings 'LEADING:20 TRAILING:20 ILLUMINACLIP:adapter.fa:2:40:15 SLIDINGWINDOW:4:20 MINLEN:36'. Reads were subsequently aligned to the *L. donovani* LdBPKv2 [21] reference genome using BWA Mem version 0.7.15 [22] with default parameters. The resulting bam files were sorted and indexed with samtools [23]. Mapping reports were generated using samtools flagstat.

Somy estimations and gene copy numbers were calculated as described in Downing (2011) [24]. Briefly, the somy value of a chromosome (S) was calculated by dividing its median sequencing depth (i.e. the number of times a nucleotide is read during sequencing, $d_{ch}$) by the median $d_{ch}$ of all 36 chromosomes ($d_{mch}$) and multiplying this value by two (for a predominantly diploid organism). Gene copy number per haploid genome (d) was defined as a raw depth for that gene ($d_r$), divided by the median depth of its chromosome ($d_{ch}$): so that $d = d_r/d_{ch}$. Somy values in our bulk-sequencing analyses are often not discrete due to mosaic aneuploidy (i.e. presence of different kayotypes among individual cells present in 1 strain) in *Leishmania*. Gene copy number estimations per cell (full depth, gene dosage) were then calculated by multiplying d with S. SNPs were previously characterized in [11]. RNA-Seq gene read count tables were generated with the HTSeq tool version 0.6.1 [25]. Count data were normalized for library size by dividing each transcript count by the geometric mean of all counts for that library [26].

## Proteomics

**Sample preparation.** Cell pellets were efficiently lysed using 200 μl RIPA buffer and 1x HALT protease inhibitor (Thermo Scientific), combined with a 30 sec sonication (Branson Sonifier SLPe ultrasonic homogenizer, Labequip, Ontario, Canada) with an amplitude of 50% on ice. After centrifugation of the samples for 15 min at 10,000 g at 4 ˚C, the cell pellet was discarded. Next, the protein concentration was determined using the Pierce BCA protein Assay kit in combination with a NanoDrop 2000 photospectrometer (Thermo Scientific).

For each sample, 25 μg of proteins were reduced using 2 μl of 200 mM tris(2-carboxyethyl) phosphine, in a volume of 20 μl 200 mM triethylammonium bicarbonate (TEAB), and incubated for 1 h at 55˚C. After alkylation of the proteins with 2μL of 375 mM iodoacetamide for 30 min protected from light, 6 volumes of ice-cold acetone were added, and the samples were incubated overnight at -20˚C. The next day, samples were centrifuged for 10 min at 10.000 g at 4˚C, the acetone was removed, and pellets were resolved in 20 μl of 200 mM TEAB. Proteins were then digested with trypsin (Promega) overnight at 37˚C with an enzyme trypsin ratio of 1:50. Before LC-MS/MS analysis, the samples were desalted with Pierce C18 spin columns according to the manufacturer's instructions (Thermo Scientific).

**Nano reverse-phase liquid chromatography and mass spectrometry.** Each of the digested samples was separated by nano reverse phase C18 (RP-C18) chromatography on an Easy-nLC 1000 system (Thermo Scientific, San Jose, CA) using an Acclaim C18 PepMap 100 column (75 μm x 2 cm, 3 μm particle size) connected to an Acclaim PepMap RSLC C18 analytical column (50 μm x 15 cm, 2 μm particle size) (Thermo Scientific, San Jose, CA). Of each sample, a total of 1μg of peptides were loaded on the column. Before loading, digests were dissolved in mobile phase A, containing 2% acetonitrile and 0.1% formic acid, at a concentration of 1μg/10μL and spiked with 20 fmol Glu-1-fibrinopeptide B (Glu-fib, Protea biosciences, Morgantown, WV). A linear gradient of mobile phase B (0.1% formic acid in 100% acetonitrile) from 0 to 45% in 90 min, followed by a steep increase to 100% mobile phase B in 10 min, was used at a flow rate of 300 nL/min. Liquid Chromatography was followed by MS (LC-MS/MS) and was performed on a Q-Exactive Plus mass spectrometer equipped with a nanospray ion source (Thermo Fisher, Waltham, MA, USA). The high-resolution mass spectrometer was set up in an MS/MS mode where a full scan spectrum (350–1850 m/z, resolution 70,000) was followed by a high energy collision activated dissociation (HCD) tandem mass spectra (100–2000 m/z, resolution 17,500). Peptide ions were selected for further interrogation by tandem MS as the twenty most intense peaks of a full scan mass spectrum. The normalized collision energy used was set at 28%. A dynamic exclusion list of 20 sec for the data-dependent acquisition was applied.

**Proteomic data analysis.** Thermo raw files were converted to mzML files using MSConvert v3.0. Label-free protein quantification (LFQ) was carried out with MaxQuant version 1.6.0.16 [27] using the following settings (other settings were kept default): Oxidation (M) and acetyl (Protein N-term) were indicated as variable modifications, carbamidomethyl (C) was indicated as a fixed modification, digestion with trypsin with maximum two missed cleavages, match between runs = yes, dependent peptides = yes. The search database was the LdBPKV2 proteome previously published by our group [12], and reversed proteins were used as decoys.

## Metabolomics

**Sample preparation.** Sample pellets were extracted using a 1:3:1 (v:v:v) chloroform/methanol/water extraction, subjected to liquid chromatography using a 2.1 mm ZIC-HILIC column (Sequant), and analyzed using the Orbitrap Exactive (Thermo Fisher Scientific) platform at the Glasgow Polyomics Center (University of Glasgow, Glasgow, Scotland) exactly as described elsewhere [28,29]. Both negative and positive ion modes were run in parallel with rapid polarity switching. Samples were analyzed in randomized order and the same analytical batch. Additionally, to aid accurate metabolite identification, verify LC-MS stability, and identify contaminants, the following control samples were included as well: (1) solvents blanks, (2) serial dilutions of a pooled sample (Undiluted, 1/2, 1/4, 1/8, and 1/16), (3) Authentic standard mixes containing in total 217 metabolites (50–400 Da) and representing a wide array of metabolic classes and pathways, (4) An amino acid standard mix (Sigma Product No. A9906).

**Metabolomic data analysis.** Data preprocessing was carried out with the XCMS 1.42.0 [30] and mzMatch 1.0.1 [31] packages in R, exactly as described elsewhere [32]. Briefly, raw spectra (mzXML format) were first subjected to retention time alignment with ObiWarp [33], after which peak detection was carried out with the centWave algorithm [34]. Peaks were then filtered: 1) Corresponding peaks from the four biological replicates were allowed a maximum reproducibility standard deviation (RSD) of 0.5, 2) Peak shape (CoDA-DW > 0.8), 3) Detection in at least 3 out of 4 replicates of a single strain and 4) Minimal peak intensity was set to 3000. The peaks were putatively identified using sequentially the LeishCyc database [35], LIPID MAPS [36], KEGG [37], the peptide database included in mzMatch, and the Human

Metabolome Database [38]. Finally, the filtered peaks were normalized for total ion count and subjected to manual quality screening.

## Integrated multi -omic analysis

Pairwise comparisons between aneuploid strains: For each comparison between two strains, we calculated the $Log_2$ fold change ($Log_2FC$) for genes (gene dosage), transcripts, proteins and metabolites. These $Log_2FC$ values were obtained by taking the $Log_2$ ratio of the average abundance between two strains. Associated *p*-values were obtained by performing a Student *t*-test for each comparison and subsequently correcting it with the Benjamini-Hochberg algorithm to limit the False Discovery Rate (FDR) to 5%. A Log2FC cutoff of 1 and adjusted *p*-value cutoff of 0.05 was used for all 'omic layers, except for gene dosage, for which only copy number differences of at least 0.5/haploid genome were considered as biologically significant [39].

Average transcript and protein abundance levels per chromosome per strain: First, the median was taken of the $Log_2$ normalized transcript counts (see respective section) and $Log_2$ LFQ protein values, per chromosome per strain. This yielded an expression value for each chromosome in each strain, for both transcript and protein expression. Then, for each chromosome, the median of the expression values of all strains that were disomic (determined based on DNA sequencing data) for that chromosome was selected as the 'disomic reference value' for that chromosome (i.e., the expression value that corresponds to the median expression of a transcript/protein on a disomic chromosome). The remaining per-strain per-chromosome expression were then divided by this disomic reference value, multiplied by 2. The code for these normalizations and analyses is available at (https://github.com/CuypersBart/Leishmania_Multiomics).

All plots were generated using the ggplot v 2_2.1.0 package [40], except the Circos plots which were created with Circos v0.69 [41]. Gene ontology enrichments were calculated with the Fisher Exact test using Python 3.8.5 and the matplotlib v3.3.4 [42] and SciPy v1.6.1 [43] libraries. GO annotations were obtained from the gene ontology consortium (http://www.geneontology.org/).

## Dosage attenuation analysis

Dosage attenuation analysis: This analysis was carried out on the only three chromosomes (Ld05, Ld08, and Ld33) that were at least once disomic (somy of 1.7–2.3), trisomic (somy of 2.7–3.3), and tetrasomic (somy of 3.5 or more) in our six study strains and had more than 10 detected proteins. The abundance values of transcripts (corrected for library size, see above) and proteins (LFQ) were divided by their abundance in the strain that was the most disomic for their encoding chromosome, according to the genome sequencing data. These expression values were then multiplied by 2 so that the normalized abundance at disomy = 2. Strains were always normalized by another strain than themselves to avoid underestimation of the standard deviation. Therefore, we do not have disomic distributions for Ld08 since Ld08 was exclusively disomic in BPK282, and thus already was used as reference strain.

Functional dosage attenuation analysis: For this analysis, we defined a protein's attenuation level as its $Log_2$ transcript fold change (i.e., its transcript level on the aneuploid chromosome divided by its transcript level on disomic chromosome), minus its $Log_2$ protein fold change. We included only 85 transcripts and proteins for this analysis, which had very accurate protein measurements (SD between biological replicates < 0.25 $Log_2FC$ units). The following protein features were checked for their relation with protein dosage attenuation: degree (i.e., number of protein interactions as transferred from STRING), length (as a proxy for transcript and protein molecular size), GRAVY (grand average of hydropathy), solubility (soluble versus

membrane protein) and subcellular location. We added transcript abundance as a covariate to the model, as the compensatory effect can be expected to become larger at higher differential transcript levels. The code these normalizations and analyses is available at (https://github.com/CuypersBart/Leishmania_Multiomics).

## Results

### Aneuploidy globally impacts transcriptome and proteome of affected chromosomes

We examined the impact of aneuploidy in *L. donovani* on transcriptome and proteome by integrated multi-omic analysis of 6 *L. donovani* strains from India and Nepal (S1A Table). These closely related strains are almost identical at the sequence level but differ greatly in their aneuploidy [44]. Specifically, the entire genome between any two strains of this set differs at most by 64 non-synonymous SNPs and one indel in coding regions (S1B Table). Thus, they are attractive model strains for studying the impact of aneuploidy on transcriptome and proteome with minimal interference of DNA sequence variation.

Genome, transcriptome, and proteome were obtained from the *in vitro* replicative promastigote stage (LOG) of the parasite. All genomes reached an average sequencing coverage of at least 26.5 fold, and 97.9% of the sequenced sites reached 10 X coverage or more. Transcript samples had on average 4.71 million reads and obtained an average mapping efficiency of 97%. Detailed mapping and coverage statistics for both genome and transcriptome data are available in S1C and S1D Table. Identified indels and estimations of chromosomal somy (i.e., average chromosome copy number in the population of cells) and dosage of each gene are available in S2 Table. Identified transcripts and their counts are available in S3 Table. Proteome analysis identified in total 3145 proteins across all samples (S4 Table).

Our 6 *L. donovani* strains showed high but varying degrees of aneuploidy (Fig 1). The most aneuploid strain was BPK288 with 12 aneuploid chromosomes, followed by BPK275 (10), BPK173 (10), BHU575 (10), BPK178 (9), and BPK282 (6). In *L. donovani*, chromosomes are numbered by increasing size. The baseline somy of chromosomes in *L. donovani* is 2N, except for chromosome 31, which is always 4N or more. Smaller chromosomes were more frequently aneuploid (Ld01, Ld05-Ld09, Ld11-Ld16), than larger chromosomes (Ld20, Ld23, Ld26, Ld31, Ld33, Ld35). Chromosome Ld31 was confirmed tetrasomic in all strains, and chromosome Ld23 always trisomic. Other aneuploid chromosomes had a variable somy. In bulk genomic analysis, somy values represent the average of the somy of individual cells present in a given population. Because of mosaicism, i.e. the presence of different karyotypes among individual cells of the parasite population in culture, many somy values were not discrete. The highest somy values (apart from Ld31) were observed for chromosomes Ld02 (somy = 3.9), Ld08 (3.9), and Ld33 (3.5) in BPK275, and for chromosome Ld05 (3.6) in BPK288.

The transcript abundance of aneuploid chromosomes mirrored almost perfectly their underlying somy (Fig 2, S1A and S1B Fig). Specifically, the average abundance of a transcript encoded by a chromosome was proportional with a factor of 0.94 with the chromosomal somy (Fig 3: 95% CI: 0.92–0.96 and $p < 2 \cdot 10^{-16}$, $R^2$: 0.98). The average abundance of the proteins encoded by a chromosome was also highly proportional to the aneuploidy of the chromosome ($p < 2 \cdot 10^{-16}$), albeit with a decreased slope of 0.76 (95% CI: 0.70–0.77, $R^2$: 0.92). Thus, aneuploidy results in a linear, but not equivalent, increase in protein abundance. Circos plots clearly showed that transcript- and protein abundance of aneuploid chromosomes was affected globally and not restricted to specific chromosomal regions (Fig 2, S1A and S1B Fig).

Interestingly, chromosome 35 exhibited only a partial amplification in strain BPK275, the first part being disomic (somy = 2.2) and the second part, starting from gene

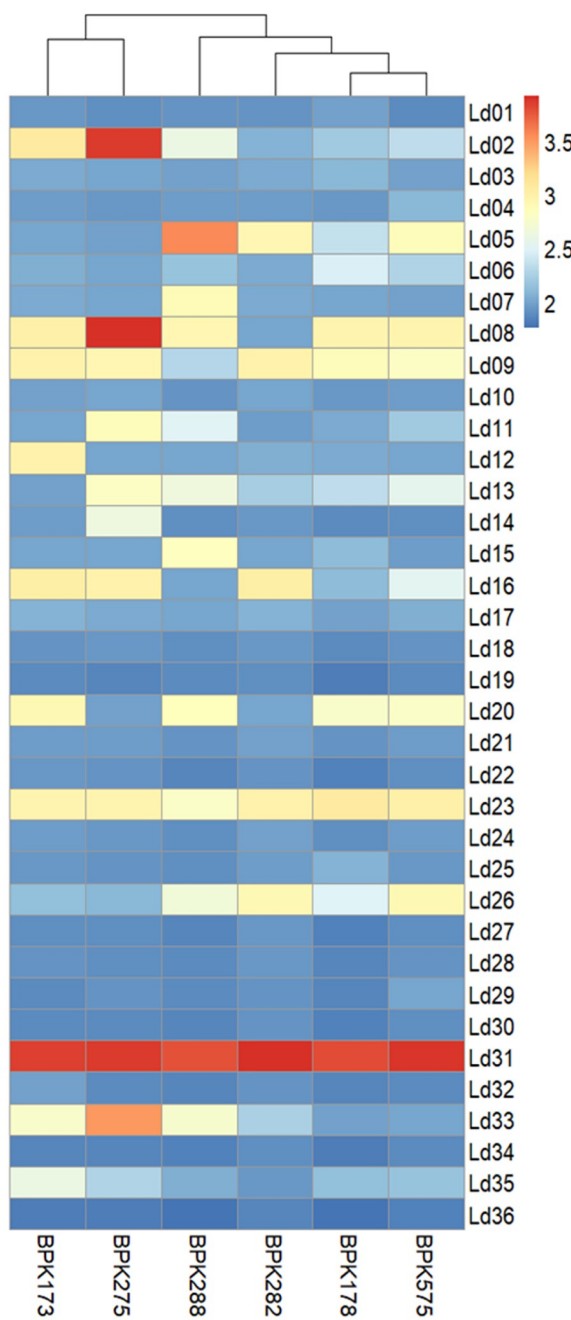

**Fig 1. Heatmap showing aneuploidy patterns in the six *L. donovani* strains.**

LdBPK_350040900, being trisomic (somy = 3.1) (Fig 2). This phenomenon of partial chromosomal amplification has previously been observed for chromosome 23 as well [45,46]. The difference in both average transcript and protein abundance between the first and the second chromosome part was significant (t-test $p$ transcript = 4.0x10$^{-82}$, protein = 7.0x10$^{-7}$), the second part having on average 1.43x more transcript abundance and 1.22x more protein abundance. This demonstrates that also partial chromosomal amplifications affect transcript and protein abundance proportionally in the amplified region.

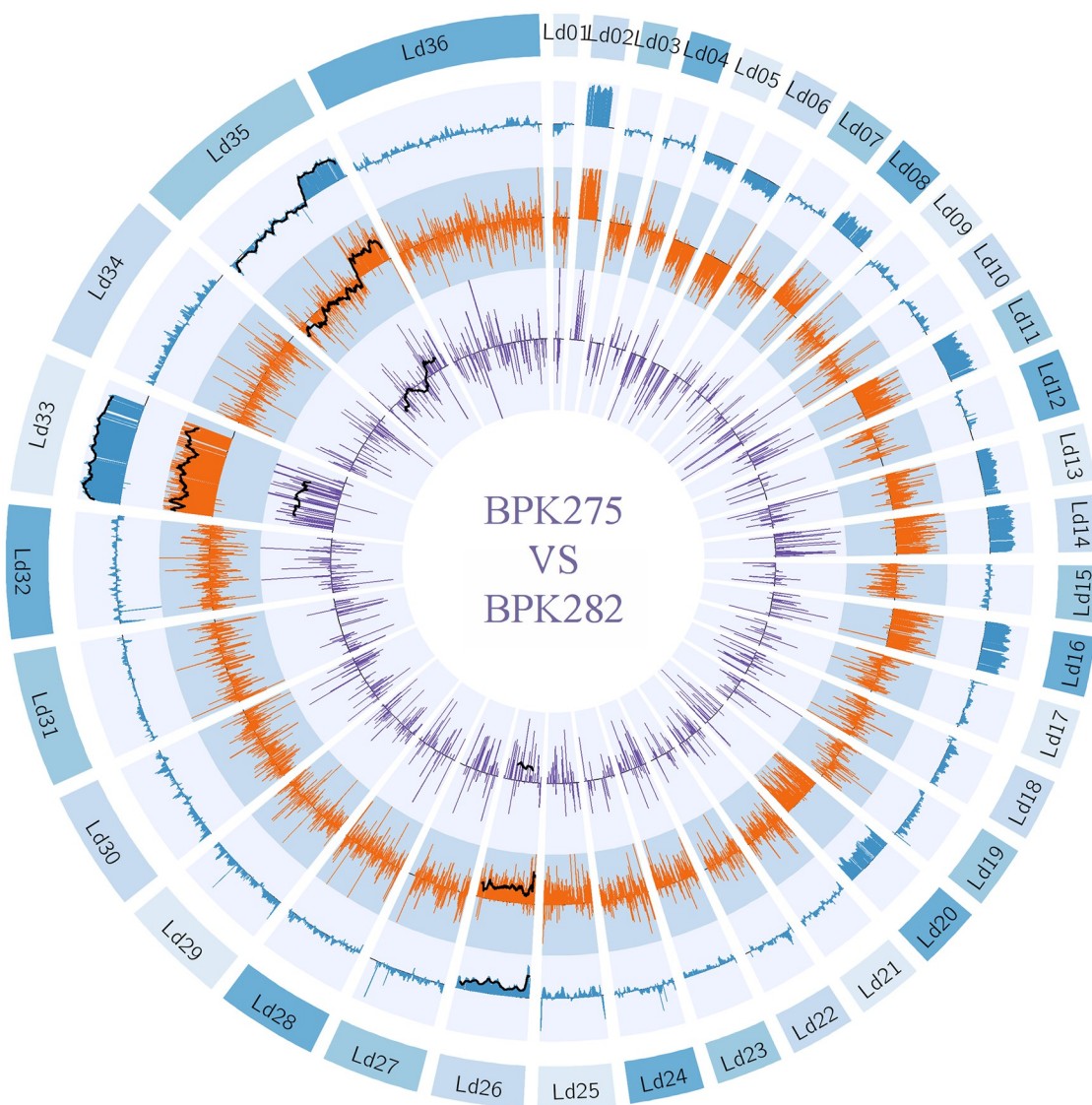

**Fig 2. Comparative Circos plot between BPK275 and BPK178, showing their relative (expressed in fold change) gene dosage (blue), transcript abundance (orange), and protein abundance (purple) across the 36 *L. donovani* chromosomes.** 20 point moving averages are indicated with a black line for the three largest aneuploid chromosomes. We show this strain pair because of the partial chromosomal amplification on chromosome Ld35. However, Circos plots between BPK173 and BPK282, and BHU575 and BPK288 show similar agreement between genome, transcriptome, and proteome and can be found in S1A and S1B Fig).

## Aneuploidy does not impact all proteins equally

Our observation that a somy increase does not lead to an equivalent increase in protein abundance, but 0.76x the somy change, suggested either of two possible explanations. A first possibility is that all proteins encoded by an aneuploid chromosome are modulated equally and exhibit only a partial protein abundance increase. We further refer to this explanation as the 'dampening hypothesis'. Such a general 'dampening' effect would suggest an underlying process related to intrinsic mechanisms or limitations in the parasite's translation system. Alternatively, different proteins could respond differently to increased gene dosage and transcript abundance, pointing to attenuation effects linked to the identity, function, and

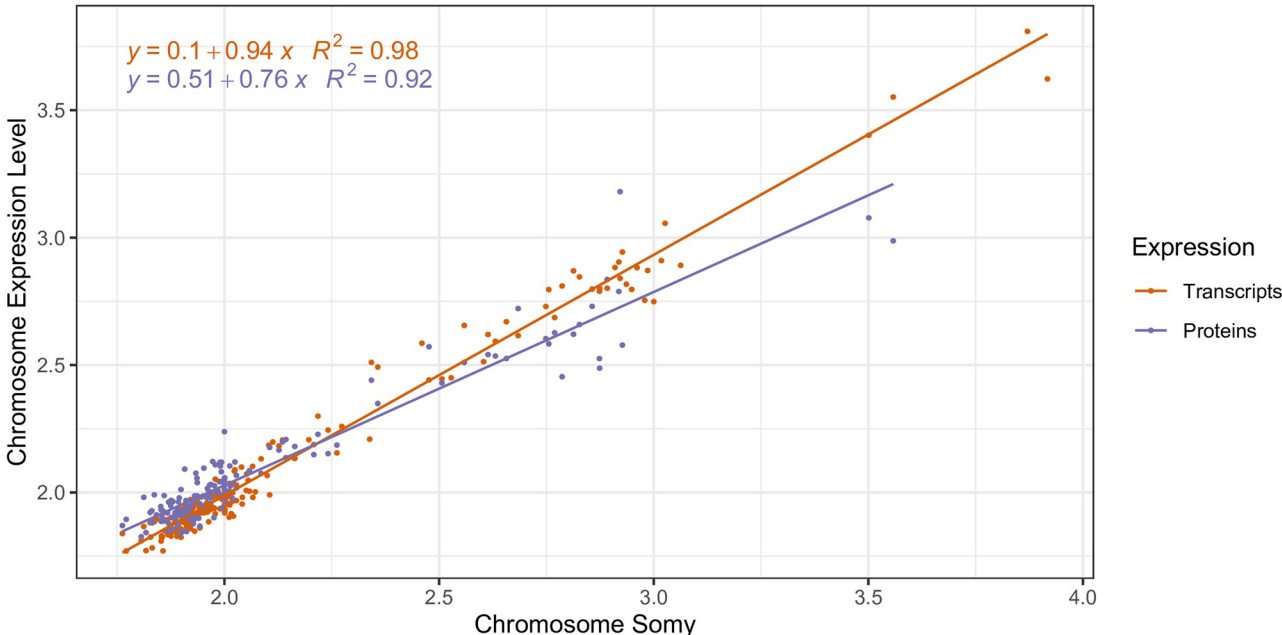

**Fig 3. Impact of chromosome aneuploidy on average chromosome transcript (red) and protein (blue) abundance levels.** Each dot represents 1 chromosome in one of the six studied aneuploid *L. donovani* strains in the replicative (LOG) phase. Abundance levels were normalized so that two transcript/protein abundance units match the abundance level of a disomic chromosome. Somy values are often not discrete due to mosaic aneuploidy (i.e. presence of different kayotypes among individual cells present in 1 strain) in *Leishmania*.

physicochemical properties of those specific proteins or certain functional protein groups. We refer to this explanation as the 'attenuated dosage response hypothesis'.

To evaluate both hypotheses, the differential abundance patterns of proteins were compared to their encoding chromosome differences from disomy to trisomy and tetrasomy. Following the 'dampening hypothesis', the average of the protein abundance distribution should increase with increased somy, but the distribution shape should remain unaltered. In our study, three chromosomes (Ld05, Ld08, and Ld33) were at least once disomic, trisomic, and tetrasomic in one of our six study strains (Fig 1), allowing for such comparative analysis. Ld02 also matched these criteria but was excluded from the analysis as only nine Ld02 proteins were detected. For reference, the same analysis was conducted for transcripts.

The average abundance of transcripts increased with somy, and the shape of the abundance distribution was found to be similar (Fig 4A). Specifically, the transcript abundance standard deviation did not significantly change with somy (Regression *p*: 0.10, Fig 4B), and the distributions were symmetric. This suggests that a somy change affects most transcripts equally, and attenuated dosage responses at the transcript level are minimal or absent. Chromosome Ld08 had a higher variance in transcript abundance than the two other chromosomes. Our data is not conclusive as to whether this is due to the shift in somy or due to an intrinsically higher transcript abundance variance for this chromosome.

The average abundance of proteins also increased with somy, but the distribution shape was not consistent. This was particularly evident when comparing the disomic versus the tetrasomic chromosomes. We observed a clear left-tailing of the protein abundance distribution on chromosomes Ld08 and Ld33 chromosomes towards the lower, more conservative fold change values. Thus, while a subset of proteins follows the aneuploidy-induced gene dosage changes, another subset is maintained at conservatively lower levels. This reduced dosage response was

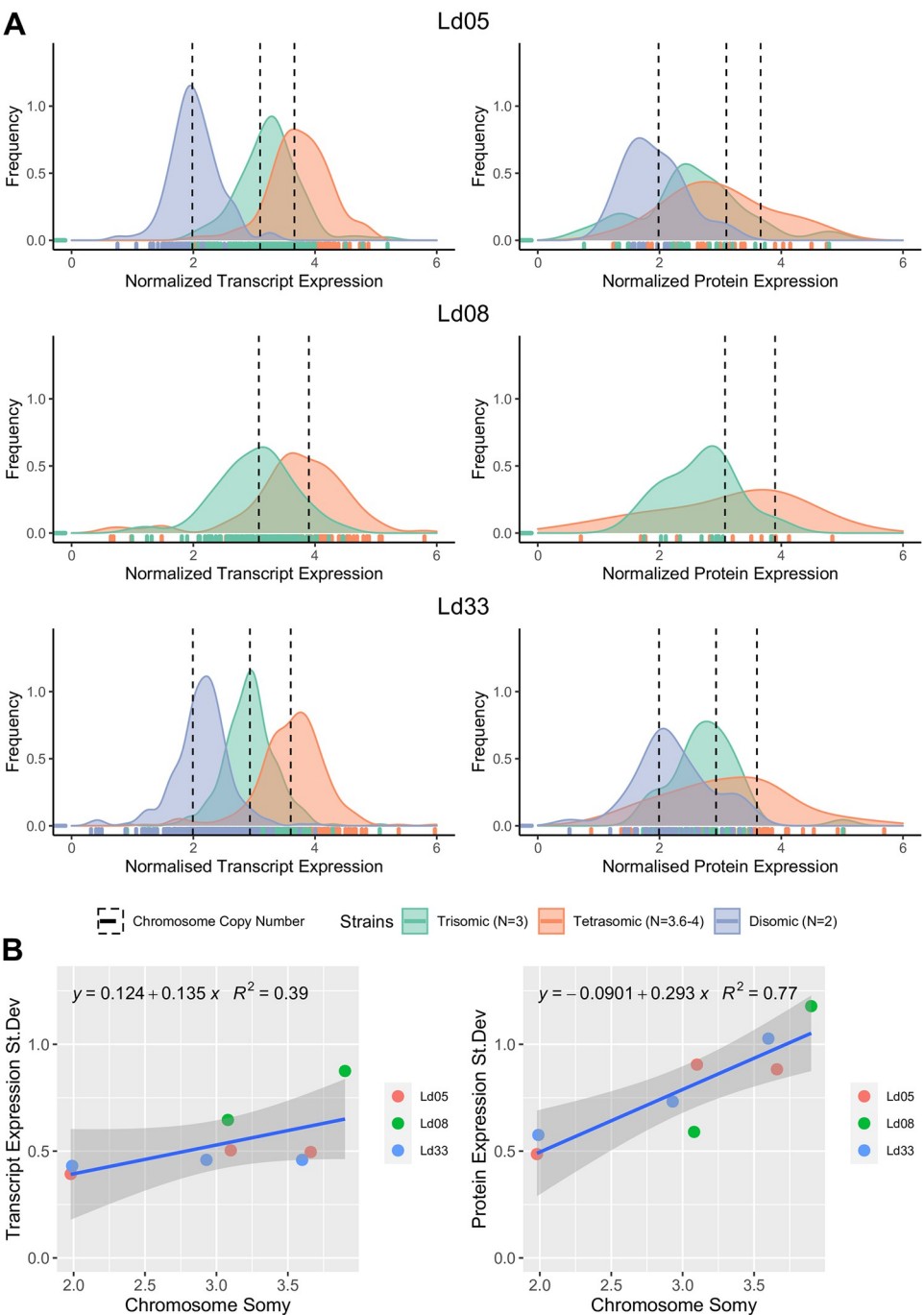

**Fig 4. A**: Normalized transcript (left) and protein (right) abundance distributions of chromosomes Ld05, Ld08, and Ld33 in their disomic (excl. chr 8), trisomic and tetrasomic states, each time observed in one of our 6 study strains. Vertical dotted lines indicate the corresponding somy, derived from the genomic data. The small vertical lines at the base of the plot indicate individual datapoints for transcripts and proteins, from which the distributions were generated. The abundance values of transcripts and proteins were normalized so that the normalized abundance at disomy = 2. This normalization was carried out with a strain that was disomic for the chromosome under investigation. We do not have disomic distributions for Ld08 since Ld08 was exclusively disomic in BPK282, and thus already was used as reference strain. **B.** Regression between the somy and transcript (left) or protein (right) standard deviation for chromosomes Ld05, Ld08, and Ld33 in different somy states.

also reflected in a general and significant increase in protein abundance variance for all three chromosomes (Regression $p$: 0.004).

## Proteins with a reduced dosage response are primarily protein complex subunits or non-cytoplasmic

We investigated if specific protein properties were associated with a reduced dosage response in our dataset. For this analysis, we defined a protein's reduced dosage effect or 'attenuation level' as its $Log_2$ transcript fold change (i.e., its transcript level on the aneuploid chromosome divided by its transcript level on disomic chromosome), minus its $Log_2$ protein fold change. A particular challenge in this analysis was that protein fold changes, and hence, the observed attenuation levels, were generally very small and close to measurement noise levels. For example, for a fully attenuated protein (to disomic levels) from a trisomic chromosome, the attenuation level would only be 0.5 $Log_2$FC units ($Log_2$FC transcript $\approx$ 0.5, $Log_2$FC protein = 0). Therefore, we focused on a set of 85 transcripts and proteins, which had very accurate protein measurements (SD between biological replicates < 0.25 $Log_2$FC units). These covered a broad range of attenuation levels ranging from 1.18 $Log_2$FC units, to values close to zero, or even some proteins that had a slight overamplification compared to the transcript level (-0.40 $Log_2$FC units) (S1E Table).

We noted that within the top 5 attenuated proteins in this high-accuracy set, 2 coded for subunits of protein complexes: i.e., the cleavage and Polyadenylation Specificity Factor (CPSF) subunit A and the dynein light chain had respectively 1.12 and 0.96 Log2FC units lower protein than transcript abundance. Accordingly, we hypothesized that protein complexes might be more likely to be attenuated ($> = 0.30$ Log2FC units). To test this systematically, all 85 proteins were screened for their involvement in heteromeric protein complexes (using annotation keywords 'subunit' and 'chain', S1E Table). Strikingly, complex subunits accounted for 15.6% (5 out of 32) of the attenuated proteins which was significantly more than the 1.8% (1 out of 53) we observed for the unattenuated proteins (Fischer exact $p = 0.0265$).

Next, we looked if there was a relation between a proteins' attenuation level and a set of biological and physicochemical protein properties to identify more subtle patterns. Specifically, we checked 5 protein properties for their relationship with protein attenuation using AN(C) OVA (S1E Table). Transcript abundance was added as a covariate to the model, as attenuation was larger at higher differential transcript levels (Adj $p = 2.4 \times 10^{-4}$). The variables 'membrane protein' (yes or no) (FDR adjusted $p = 0.696$), the number of protein interactions (Adj. $p = 1$), protein length (a proxy for transcript and protein size) (Adj. $p = 1$) and hydrophobicity (Adj. $p = 1$) had no significant relationship with a protein's attenuation level. In contrast, the subcellular location (Adj. $p = 0.032$) of a protein was significantly linked to its degree of dosage attenuation. To investigate this further, we investigated the relation between each subcellular compartment and the attenuation level of its proteins (S1E Table). Cytoplasmatic proteins (n = 39) were significantly less attenuated compared to all other proteins (Adj. $p = 0.011$). Thus, cytoplasmatic proteins follow aneuploidy-induced dosage changes more closely than proteins from other cellular compartments. In contrast, proteins associated with the Golgi apparatus (n = 2) had a significantly higher attenuation levels than all other proteins (Adj. $p = 0.0065$). However, this result needs to be interpreted with caution as the sample size was only 2 proteins. These two proteins were a vesicle-associated membrane protein and a small GTP-binding protein Rab18, both known to be secreted. Proteins from mitochondria (Adj. $p = 0.32$, n = 18), nucleus (Adj. $p = 0.50$, n = 13), peroxisome (Adj. $p = 0.81$, n = 6) and ER (Adj. $p = 0.74$, n = 2) did not have significantly attenuation levels compared to proteins from all other compartments.

## The impact of aneuploid chromosomes on the gene products of their euploid counterparts

In the previous sections, we reported how the aneuploidy of a chromosome affects the abundance of its encoded transcripts and proteins. However, through perturbation of regulatory networks, these changes might also affect transcripts and proteins encoded by genomic regions unaffected by aneuploidy. These were previously defined as 'trans-effects' and have been observed on the transcriptional level in diverse species, including *Arabidopsis* and *Drosophila*, as well as in humans with sex chromosome aneuploidies, Turner or Klinefelter syndrome [47–50]. Here we investigated the extent of trans-effects in *Leishmania* by comparing and characterizing the fractions of differential transcripts and proteins coming from aneuploid chromosomes, i.e. primary or cis-transcripts and cis-proteins, versus those coming from euploid chromosomes, i.e. trans-transcripts and trans-proteins.

We carried out differential transcript and protein abundance analyses (FDR adjusted $p < 0.05$) between all possible pairs of the six *L. donovani* study strains (Table 1). On average, 71.8% of differentially expressed transcripts were cis-transcripts, while cis-proteins constituted only 40.7%. The remaining 28.2% of differentially expressed transcripts and 59.3% of proteins were thus encoded by euploid chromosomes (trans-) and have therefore been regulated by another mechanism than gene dosage. Interestingly, only 20.0% of these trans-proteins had a corresponding transcript change ($|Log2FC| > 0.5$), indicating the vast majority is regulated at protein level directly and not through modulation of mRNA levels. Importantly, none of these trans-transcripts nor trans-proteins were associated with local copy number variants: these were rare and excluded from this analysis. Also sequence variation between strain pairs is unlikely to have resulted in the observed proportions of differential trans-transcripts and trans-proteins. Only 11 trans-transcripts contained SNPs between our strains (6 missense, 1 synonymous and 4 upstream SNPs) and 2 proteins (1 upstream and 1 downstream SNP) (S1G and S1H Table). Therefore, our hypothesis is that aneuploidy induces direct changes to cis-transcripts and cis-proteins, which directly (e.g., protein-protein interaction) or indirectly drive abundance changes in trans-transcripts and mostly, trans-proteins.

Next, we looked deeper into the functions of trans-transcripts and proteins. After pooling the trans-transcripts and trans-proteins from all comparisons, we identified a set of 541 unique trans-transcripts and 203 unique trans-proteins (duplicates and hypothetical proteins

**Table 1. Total numbers (N) of differential (FDR adjusted $p < 0.05$) transcripts and proteins between all possible pairs of our 6 *L.donovani* strains.** Each time, it is indicated what proportion of these transcripts or proteins originate from chromosomes that have somy changes (because of aneuploidy) in the strain pair (Cis) or chromosomes that have identical somies between the two strains (Trans). Five comparisons are not shown because they contained less than 15 differential transcripts or proteins. These are shown in S1F Table.

| Strain Comparison | Differential Transcripts | | | Differential Proteins | | |
|---|---|---|---|---|---|---|
| | Cis (%) | Trans (%) | N | Cis (%) | Trans (%) | N |
| 275vs178 | 79.8 | 20.2 | 386 | 44.8 | 55.2 | 105 |
| 275vs173 | 86.6 | 13.4 | 134 | 40.7 | 59.3 | 91 |
| 275vs282 | 77.6 | 22.4 | 67 | 46.4 | 53.6 | 84 |
| 288vs282 | 60.9 | 39.1 | 23 | 39.3 | 60.7 | 61 |
| 275vs575 | 84.8 | 15.2 | 145 | 58.9 | 41.1 | 56 |
| 173vs288 | 86.3 | 13.8 | 80 | 55.6 | 44.4 | 27 |
| 173vs282 | 74.4 | 25.6 | 39 | 48.1 | 51.9 | 27 |
| 173vs178 | 55.0 | 45.0 | 40 | 26.3 | 73.7 | 19 |
| 575vs282 | 41.2 | 58.8 | 34 | 6.3 | 93.8 | 16 |
| Average | 71.8 | 28.2 | | 40.7 | 59.3 | |

removed, S1G and S1H Table). GO-overrepresentation analysis showed that no GO terms were significantly enriched in the pooled list of trans-transcripts or trans-proteins from all comparisons (FDR adjusted *p* threshold 0.05). However, it must be noted here that GO- and other annotations in *Leishmania* are often relatively unspecific due to the parasite's distinct evolutionary position, and therefore low gene homology with better studied model organisms. For example, only 48.7% of the *L. donovani* genes could be annotated with at least one GO term, and even these are often of the highest hierarchies (low specificity)[12]. As an alternative enrichment analysis approach using the full genomic annotations directly, we first manually checked if any frequent keywords could be observed occurring in annotations of the trans-proteins and identified 10. We then compared if these keywords were significantly enriched versus all detected proteins with a Fisher Exact test. 7 of these were indeed significantly enriched of which 3 related to protein metabolism: peptidase activity (FDR adjusted $p = 9.4 \times 10^{-5}$), heat-shock protein (Adj. $p = 2.3 \times 10^{-2}$) and chaperon(in) (Adj. $p = 2.1 \times 10^{-2}$), 2 related to metabolism: glutamate (Adj. $p = 1.9 \times 10^{-2}$) and long-chain-fatty-acid-CoA ligase (Adj. $p = 5.4 \times 10^{-3}$), and two others: mitochondrial (Adj. $p = 3.0 \times 10^{-2}$) and hypothetical protein (Adj. $p = 3.4 \times 10^{-7}$). At RNA-level none of the keywords were significant. This demonstrates that differential trans-proteins associated with aneuploidy are not a random occurrence, but related to specific functional classes and processes. Particularly relevant in this context are the enrichment of heat-shock proteins and chaperones, as aneuploidy is known to burden protein folding pathways [51].

## Aneuploidy globally affects transcriptome and proteome regardless of life stage

So far, this study discussed results from the logarithmic growth stage (LOG), which essentially consists of *Leishmania* procyclic promastigotes (replicating, non-infectious). However, despite the absence of transcriptional regulation of individual protein-coding genes, transcripts and proteins can undergo post-transcriptional regulation during parasite differentiation. This raised the question whether the correlation between chromosome somy, average transcript abundance, and average protein abundance is also present in other life stages. Hence, for three strains, we compared the LOG growth with the STAT growth phase. STAT is highly enriched for metacyclic promastigotes, which are non-replicating, have a distinct morphology, are metabolically different, and infectious to the human host [52]. This was validated with metacyclogenesis markers META1 and HASPB, which were significantly upregulated in STAT as they are expected to be (Adj. $p < 10^{-13}$, S1H Table). Transcriptome and proteome measurements and analyses were carried out exactly as for the LOG phase.

For both LOG and STAT, a chromosome's somy was proportional to its average transcript expression level, and protein expression level (Fig 5, $p < 2 * 10^{-16}$ for transcript and protein in both LOG and STAT). A reduced effect of somy on protein abundance was apparent at the protein level, in both LOG and STAT growth phases. This suggests that the gene dosage effects associated with aneuploidy remain present throughout the parasite's life cycle.

## Aneuploidy results in metabolic differences between *L. donovani* strains

We ultimately aimed to understand if aneuploidy in *L. donovani* is reflected in the parasite's metabolome. Therefore, we carried out LC-MS metabolomics and confidently identified 78 metabolites across the six aneuploid study strains (S5 Table). In contrast to transcripts and proteins, there is no straightforward relation between a metabolite and a specific gene. Genes might affect metabolites directly (e.g. an enzyme), indirectly, affect multiple metabolites, or

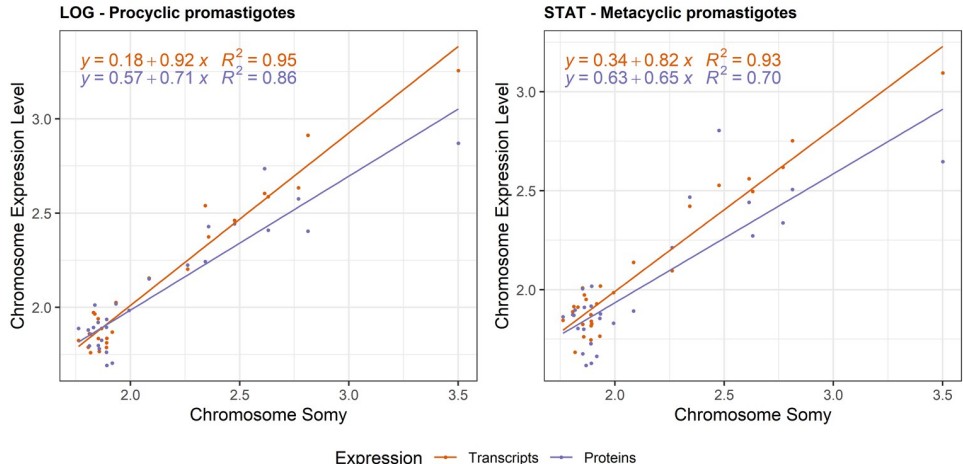

**Fig 5. Impact of chromosome aneuploidy on average chromosome transcript (red) and protein (blue) output levels in LOG phase (A) and STAT phase (B).** Each dot represents 1 chromosome in one of the 3 studied aneuploid *L. donovani* strains (BPK275, BPK178, and BPK173).

affect no metabolite at all. Therefore, we turned to an indirect strategy to integrate the metabolomic data with our previous observations.

We determined if differences in aneuploidy between strains in LOG correlate with their metabolic differences. As such, we calculated the pairwise genomic (somy), transcriptomic, proteomic, and metabolic Euclidean distance (based on all measured chromosomes, transcripts, proteins and metabolites) between each possible strain pair, and correlated these distances (Fig 6). The larger the distance between these 2 strains, the more different their respective genomes, transcriptomes, proteomes, or metabolomes are from one another. Consistent with our other analyses, we found strong correlations between the genomic and transcriptomic distance, between the transcriptomic and proteomic distance, and between the genomic and proteomic distance ($p \leq 0.001$ for all comparisons). However, the metabolome initially did not show any correlation with genome, transcriptome, and proteome. We suspected this might be caused by the high intrinsic levels of random variation of a subset of metabolites (by either technical or biological causes). Therefore, in the second phase of the analysis, we filtered on metabolites that had at least once a significant fold change (adjusted $p < 0.05$) between any pair of two strains. This strategy selects only metabolites that are stable enough to reflect true differences in metabolite levels between strains. The resulting 'differentiating metabolome' consisting of 34 metabolites correlated strongly with all omic layers ($p < 0.05$, Fig 6). This demonstrates that the degree of karyotype difference between two strains correlates with their degree of metabolic differences. Specifically, the differentiating metabolome included 9 amino acids (alanine, aspartate, cysteine, leucine/isoleucine (not distinguishable), methionine, phenylalanine, tryptophan, tyrosine, and valine), suggesting the amino acid metabolism is an important molecular difference between these aneuploid parasites.

## Discussion

*Leishmania* displays a remarkably high tolerance for aneuploidy, with any chromosome having the capacity to become polysomic and produce a viable parasite. The high karyotype dynamicity and the drastic karyotype changes between experimental conditions stress the central importance of aneuploidy for the parasite to adapt to novel environments [11]. However, it

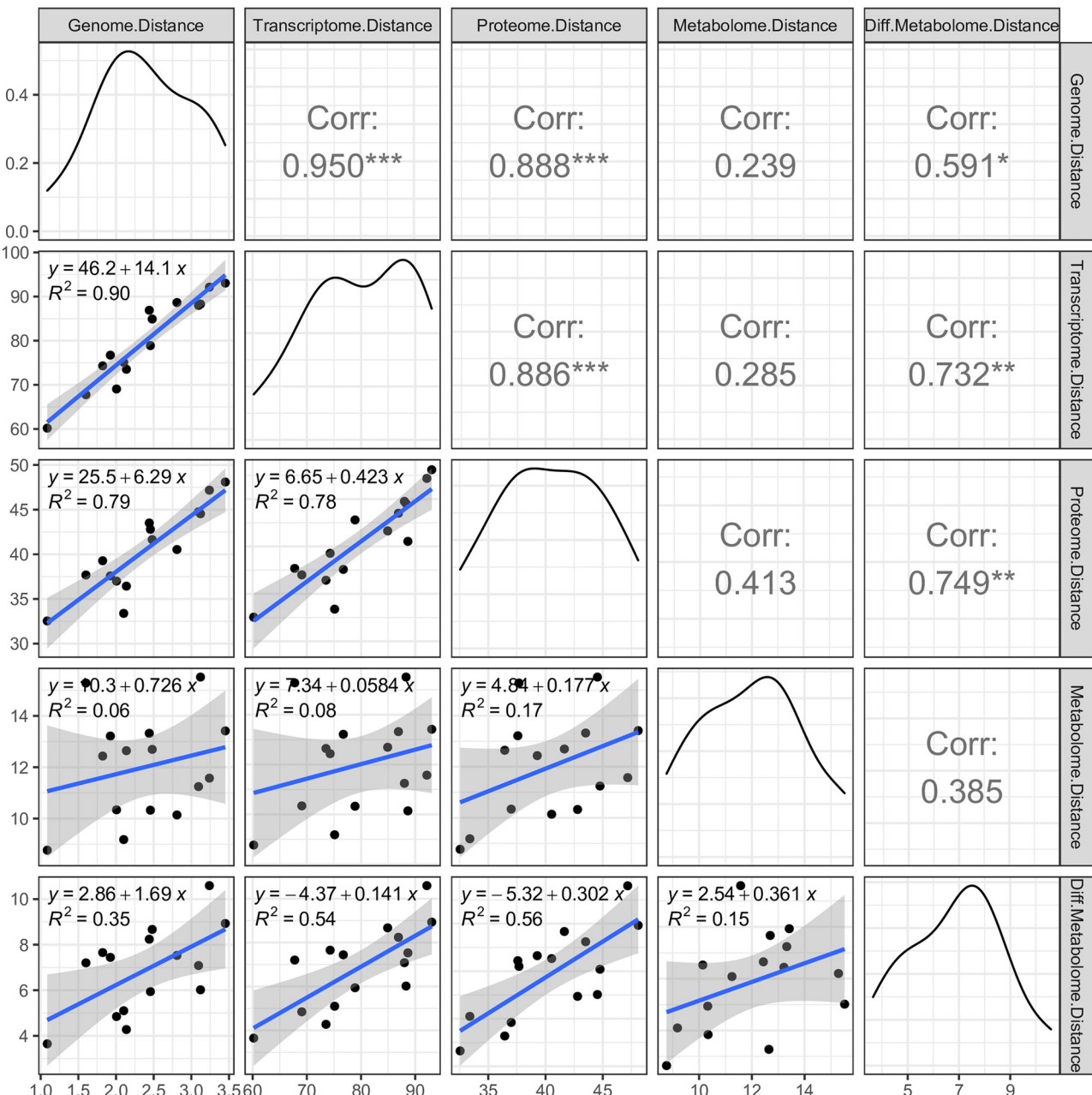

**Fig 6. How differences between two *L. donovani* strains on 1 'omic layer correlate with their differences on another omic layer.** The plots show the pairwise correlations between the genome distance (aneuploidy), transcriptome distance, proteome distance, metabolome distance (all 78 metabolites), and differentiating metabolome distance (34 metabolites), each time calculated between 2 *L. donovani* strains. Each dot represents 1 comparison between two strains. With 6 strains in our experiment, there were 15 possible pairwise comparisons. The distance metric that was used is the Euclidean distance. Somy values were used directly, but transcript abundance, protein abundance or metabolite abundance were first Z-scored before calculating the distance. The top-right panels show the Pearson correlation coefficients with * = Pval <0.05, ** Pval <0.01, *** = Pval <0.001, while the bottom-left panels show the linear regressions.

remained unclear how this large genomic unstability affects the parasite as a system. Moreover, *Leishmania*'s distinct phylogenetic status and unique gene expression system disallows extrapolation from model systems. Therefore, we investigated the systemic impact of aneuploidy in *L. donovani* by integrating, for the first time, genome, transcriptome, proteome, and

metabolome profiling data of highly aneuploid strains. Our study offers a comprehensive picture of the molecular changes associated with aneuploidy in this unique Eukaryotic model that lacks transcriptional regulation of individual protein-coding genes.

We confirmed that transcript abundance is affected across the entirety of aneuploid chromosomes. This observation is in line with previous work that has also found a strong link between aneuploidy-induced gene dosage changes and transcript abundance in *Leishmania* [12,17,53]. We further showed that this relation is almost equivalent, with transcription levels closely mirroring chromosome copy number changes (transcript abundance = 0.94x the somy). It is important to note here that to establish this 0.94x equivalence, we compared the same chromosome to its own disomic state as the reference transcript expression level at disomy. This does not exclude that chromosomes (or, on a smaller scale, cistrons) can have inherently different transcription levels. Indeed, it has previously been shown that chromosome 31 has the expression of a disomic chromosome in its default tetratomic state[17]. The impact of aneuploidy on the *L. donovani* proteome was also linear and chromosome-wide, but not equivalent. For every chromosome copy, the protein abundance levels of that chromosome increased (on average) by a factor 0.76x the somy.

This reduced impact of gene dosage on the protein level was due to the attenuation of a subset of proteins. While many proteins did follow the aneuploidy-induced gene dosage changes, a group of proteins showed a reduced dosage response. This reduction ranged from partial to complete attenuation to disomic levels. We made two observations concerning the identity of attenuated proteins that might explain this attenuation. Firstly, attenuated proteins were enriched for subunits of macromolecular protein complexes (15.6% versus 1.8% in the background set). This observation matches several observations in aneuploid yeast and human (including cancer) cell lines [54–59] that also detected protein complex subunits to be highly prevalent amongst attenuated proteins. It can be explained by the fact that individual subunits of macromolecular complexes are typically unstable unless assembled into a stable complex [57]. As subunits of a single complex are often encoded on different chromosomes and produced in balanced, stoichiometric amounts, an increase in an individual subunit (here: by aneuploidy) will often not lead to more complex formation [55,57,60]. Instead, the protein will not have an available binding partner(s), remains unstable, and rapidly degrades. It is possible that this stochiometric imbalance caused by aneuploidy is to some extent detrimental to the parasite. One argument in this direction is that we and previous studies found smaller *Leishmania* chromosomes (with smaller numbers of genes) to be more frequently aneuploid, which matches observation in yeast and humans (chromosome 21 has the smallest number of genes in humans and the least severe autosomal aneuploid) [11,24,61]. Additionally, it has been shown in yeast that genes that are deleterious when overexpressed are enriched for protein complex subunits [62]. From this perspective, the situation in *Leishmania* could be similar to other single-celled organisms such as yeast, where the fitness benefits of aneuploidy must outweigh the fitness costs in a given environment [62].

Secondly, we observed a correlation between the degree of attenuation and the cellular destination of the attenuated protein. Two secreted proteins associated with the Golgi apparatus were the most attenuated (i.e. their expression was close to disomic levels), which could be explained by the fact that secreted proteins do not accumulate in the cell. Correspondingly, attenuation of secreted proteins was also observed in a recent study of human fibroblasts with trisomy 21 [57]. Proteins from mitochondria, the nucleus, and the peroxisome were also significantly more attenuated than cytoplasmatic proteins, which had the lowest level of attenuation (i.e., cytoplasmic proteins followed gene dosage the best). A potential explanation might be that mitochondrial, nucleic, and peroxisome proteins require active transport into their destined organelle. This transport requires extensive interaction with other proteins,

potentially explaining why their abundance scales less with increased protein production. In aneuploid (disomic II) strains of *S. cerevisiae*, increased translation was found of genes related to protein trafficking pathways of the endomembrane system and nucleus, also suggesting dependency on this system [63].

Partial chromosomal amplification can occur in *Leishmania*. We have previously observed this phenomenon for chromosome Ld23 [45,46] and in the present study for Ld35. Interestingly, gene dosage proportionally affected transcript and protein abundance for these subchromosomal amplification as well. This suggests that our findings can be extrapolated to segmental aneuploidy as well.

Further, we demonstrated that these gene dosage changes remain important throughout at least several life stages of the parasite. Indeed, we observed an similar impact of gene dosage on transcriptome and proteome in either LOG (enriched for procyclic promastigotes, proliferative stage) or STAT (enriched for metacyclic promastigotes, infectious stage) growth phases. This is striking, as the parasite featuring extensive transcriptome and proteome changes during its life cycle. For example, transcriptomics studies have shown that up to 30–50% of transcripts are differentially expressed between procyclic and metacyclic promastigotes [16,64]. Similarly, 10–40% of proteins are differentially expressed between the parasite's promastigote and amastigote life stages [12,65].

A limited proportion of differential transcripts between aneuploid strains (23.6%) and the majority of differential proteins (56.9%) originated from chromosomes without a change in somy, so-called trans-transcripts, and trans-proteins. As we excluded genes with CNVS between strains from our analysis and SNPs and INDELs were very rare, we hypothesize that the observed trans-effects could result from aneuploidy, although our study does not allow to test for a causal relationship. High abundances of trans-transcripts have been reported in many other aneuploidy studies, including on *Drosophila* and *Arabidopsis*, as well as in humans with sex chromosome aneuploidies, Turner or Klinefelter syndrome [47–50]. These trans-transcriptomes were typically enriched for transcription factors, genes related to metabolic processes, protein metabolism, or cellular stress responses. It supports the idea that the cellular system mitigates the primary impact of aneuploidy by inducing transcriptional changes to trans-genes. Here, in *Leishmania*, we found relatively low proportions of differential trans-transcripts and no significant enrichment of specific functional classes amongst these. While *Leishmania* lacks transcriptional regulation of individual protein-coding genes, a more extensive regulation would have been a theoretical possibility, as post-transcriptional regulation of individual transcripts (eg. by alba-domain proteins) was observed in the context of parasite development [66]. Instead, we did find a large proportion of trans-proteins. It is possible that the abundances of these trans-proteins are modulated at the translation or protein-stability level directly, as the majority did not have underlying transcript-level changes. Interestingly, these trans-proteins were significantly enriched for specific protein groups, including protein metabolism-related chaperones and chaperonins, peptidases, and heat-shock proteins. This is relevant as aneuploidy is known to put an extra burden on the protein-folding system [51].

Finally, we showed that the degree of aneuploidy differences between strains matched the degree of transcriptome, proteome, and differential metabolome changes. This is an important observation as it suggests that aneuploidy in *Leishmania* has the capacity to drive metabolic variation, which is closely linked to the phenotype. It also matches our previous observations where we linked *L. donovani* gene copy number variants directly to changes in metabolic pathways [32]. Ultimately, it supports the view that aneuploidy can be adaptive and drive the metabolic changes that the parasite needs to survive in a novel environment.

## Conclusion

In summary, our study shows that aneuploidy in *Leishmania* globally and proportionally impacts the transcriptome and proteome of affected chromosomes throughout the parasite's life cycle *in vitro*. In *Leishmania* strains with a closely related genetic background, the degree of these aneuploidy-induced dose changes ultimately correlates with the degree of metabolomic difference. This further supports the view that aneuploidy in *Leishmania* can be adaptive. As in other Eukaryotes, we observed reduced dosage effects at protein level for protein-complex sub-units and secreted proteins. Additionally, we found non-cytoplasmic proteins to respond less to dosage changes than cytoplasmic ones. Differentially expressed transcripts and proteins between aneuploid *Leishmania* strains also originated from non-aneuploid chromosomes. At protein level, these were enriched for proteins involved in protein-metabolism, including chaperones and chaperonins, peptidases, and heat-shock proteins. We believe that the high karyotype diversity *in vitro* and absence of classical transcriptional regulation make *Leishmania* an attractive model to study processes of protein homeostasis in the context of aneuploidy and beyond.

## Supporting information

**S1 Table. S1A Table.** Detailed description of the *L. donovani* isolates used in this study. **S1B Table.** Number of total SNPs, non-synonmous SNPs and homozygous non-synonymous SNPs between any pair of *L. donovani* strains in this study. **S1C Table.** Sequencing quality and mapping statistics of all 6 sequenced *L. donovani* genomes in this study. **S1D Table**. Sequencing quality and mapping statistics of all sequenced *L. donovani* transcriptomes in this study. **S1E Table.** Protein attenuation model data, coefficients and *p*-values. **S1F Table.** Summary of all cis- and trans- transcripts and proteins of all possible pairwise comparisons between the 6 *L. donovani* strains in this study. **S1G Table.** Pooled trans-transcripts from all possible pairwise comparisons between the 6 *L. donovani* strains in this study. **S1H Table.** Pooled trans-proteins from all possible pairwise comparisons between the 6 *L. donovani* strains in this study. **S1I Table.** Upregulated metacyclogenis markers in STAT growth phase. Output generated by DESeq2.
(XLSX)

**S2 Table. Structure (Somy and local CNVs) of all sequenced *L. donovani* genomes in this study.**
(XLSX)

**S3 Table. Differential RNA abundance analyses of the different biological comparisons made in this study.**
(XLSX)

**S4 Table. Detailed statistics about all proteins detected in this study.** Output generated by MaxQuant.
(XLSX)

**S5 Table. Detailed statistics about all metabolites detected in this study.**
(XLSX)

**S1 Fig. S1A Fig and S1B Fig.** Comparative Circos plot between BPK173 and BPK288 (S1A) and between BHU575 and BPK282 (S1B), showing their relative (expressed in fold change) gene dosage (blue), transcript abundance (orange), and protein abundance (purple) across the 36 *L. donovani* chromosomes.
(DOCX)

## Acknowledgments

We thank the Center of Medical Genetics at the University of Antwerp for hosting the NGS facility on which our RNA-Seq experiments were performed. The computational resources used for this work were provided by the VSC (Flemish Supercomputer Center) at the University of Antwerp.

## Author Contributions

**Conceptualization:** Bart Cuypers, Pieter Meysman, Cedric Notredame, Jean-Claude Dujardin, Malgorzata A. Domagalska, Kris Laukens.

**Data curation:** Bart Cuypers.

**Formal analysis:** Bart Cuypers, Pieter Meysman.

**Funding acquisition:** Bart Cuypers, Jean-Claude Dujardin, Kris Laukens.

**Investigation:** Bart Cuypers, Pieter Meysman, Ionas Erb, Wout Bittremieux, Dirk Valkenborg, Geert Baggerman, Inge Mertens, Cedric Notredame, Jean-Claude Dujardin, Malgorzata A. Domagalska, Kris Laukens.

**Methodology:** Bart Cuypers, Pieter Meysman, Ionas Erb, Wout Bittremieux, Dirk Valkenborg, Geert Baggerman, Inge Mertens, Cedric Notredame, Jean-Claude Dujardin, Malgorzata A. Domagalska, Kris Laukens.

**Resources:** Dirk Valkenborg, Geert Baggerman, Inge Mertens, Shyam Sundar, Basudha Khanal.

**Supervision:** Pieter Meysman, Jean-Claude Dujardin, Malgorzata A. Domagalska, Kris Laukens.

**Visualization:** Bart Cuypers.

**Writing – original draft:** Bart Cuypers, Jean-Claude Dujardin, Malgorzata A. Domagalska, Kris Laukens.

**Writing – review & editing:** Bart Cuypers, Pieter Meysman, Ionas Erb, Wout Bittremieux, Dirk Valkenborg, Geert Baggerman, Inge Mertens, Shyam Sundar, Basudha Khanal, Cedric Notredame, Jean-Claude Dujardin, Malgorzata A. Domagalska, Kris Laukens.

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
