## [Decision Letter · Decision Letter 0]

26 Jul 2022

Dear Dr Cuypers,

Thank you very much for submitting your manuscript "Four layer multi-omics reveals molecular responses to aneuploidy in Leishmania" for consideration at PLOS Pathogens. As with all papers reviewed by the journal, your manuscript was reviewed by members of the editorial board and by several independent reviewers. The reviewers appreciated the attention to an important topic. Based on the reviews, we are likely to accept this manuscript for publication, providing that you modify the manuscript according to the review recommendations.

The consensus from Reviewer 2 and 3 is that you are over-emphasizing the novelty of the data and this isn't necessary. Please read the comments from these 2 reviewers carefully and modify your paper as required. No further experimentation is required, but the paper will be sent back to reviewers 1 and 2 for final comments.

Sincerely,

Christian R. Engwerda

Section Editor

PLOS Pathogens

Christian Engwerda

Section Editor

PLOS Pathogens

Kasturi Haldar

Editor-in-Chief

PLOS Pathogens

orcid.org/0000-0001-5065-158X

Michael Malim

Editor-in-Chief

PLOS Pathogens

orcid.org/0000-0002-7699-2064

The consensus from Reviewer 2 and 3 is that you are over-emphasizing the novelty of the data and this isn't necessary. Please read the comments from these 2 reviewers carefully and modify your paper as required. No further experimentation is required, but the paper will be sent back to reviewers 1 and 2 for final comments.

Reviewer Comments (if any, and for reference):

Reviewer's Responses to Questions

**Part I - Summary**

Reviewer #1: The revisions have adequately addressed my concerns, and I support publication of this manuscript in its current form.

Reviewer #2: This manuscript reveals that Leischmania can adapt to new environments via gene dosage effects. This can occur because the species has a large number of chromosomes that divide the genome into many segments and because of its constitutive mode of gene expression. This is an interesting case and worthy of publication following some revisions.

Reviewer #3: The authors addressed all my comments and explained well their reasoning in most cases, particularly the method details, with one notable exception. In the paragraph clarifying the novelty for eukaryotes, I am still not sure whether I understand their interpretation of their data and the comparison to other eukaryotes. Below is the text what the authors now included in discussion. They state that most aneuploids display regulation of trans-transcripts, while Lieshmania execute most effects via regulation of proteins in trans. The authors first need to clearly define the terms trans-trancript and trans-protein. Further, the difference is not all that clear. The current model is rather that the proteotoxic stress impairs many aspects of cellular physiology, and in reaction to that specific stress response pathways – autophagy, unfolded protein response, inflammatory response etc – are activated. Not all of these responses occur via activation of transcription. Some proteins get stabilized or activated (eg. proteasome proteins, autophagy), others degraded (replication factors). Thus, I think the authors should attempt a more differentiated account of the response to aneuploidy. Finally, as already stated before, why is it such a surprise that Leishmania differentially regulates the proteins and not the transcripts in response to aneuploidy? The authors explain already in the beginning of the manuscript that this is the way how Leishmania regulates gene expression. Leishmania just simply cannot do it differently.

By the way, I also believe that Leishmania is a very attractive model, but some of the arguments used here are not entirely clear and correct.

Below is the paragraph in question, with some remarks and underscored unclear

Novelty for other Eukaryotes:

Conclusion L593-L602: “While most aneuploid Eukaryotes display extensive regulation of trans-transcripts to mediate the effects of aneuploidy, we did not observe this to the same extend in Leishmania (the authors do not show any comparisons of the “extends”). This suggests that the post-transcriptional mRNA regulation system of the parasite does not mediate the impact of aneuploidy (to my knowledge post-transcriptional regulation of mRNA – do they mean mRNA stability? - has not be observed in any other aneuploids). Instead, Leishmania features a surprisingly high number of trans-proteins, which are likely post-transcriptionally modulated. It thus seems that aneuploidy is accompanied by trans-proteomic changes in Eukaryotes, regardless of the system of regulation. Post-transcriptional modulation of trans-proteins (I am sorry but I am really not sure what this means) may also be present in other Eukaryotes, but could be overshadowed by the transcription factor-induced gene regulation that is absent in Leishmania (Do the authors mean that it is impossible to recognize whether protein abundance was regulated on transcript or protein level? That’s incorrect. This data is available for other organisms and have been analyzed). For this reason, we believe that Leishmania is an attractive model to study processes of posttranscriptional modulation of protein levels in the context of aneuploidy and beyond.”

**Part II – Major Issues: Key Experiments Required for Acceptance**

Reviewer #1: No major issues

Reviewer #2: While this is a revised version from a previous review, there are some aspects that need further revision. In some cases, the authors did not seem to assimilate the suggestions about underlying principles in general and just Band Aided over some issues. In some aspects the authors might be trying too hard to claim novelty, which is not really necessary—the findings are interesting without embellishment. The Abstract and the text states that dosage compensation was found for protein complex subunits and non-cytoplasmic proteins. However, only 5 out of 32 analyzed show no change compared to the normal. Thus, this is a stretch. The basis of this is probably trivial: excess subunits for complexes are just degraded. There is no evidence provided for an active process of dosage compensation. The authors need to remove the term dosage compensation from the manuscript and instead refer to the situation as protein subunit homeostasis or something else but not dosage compensation. A claim of some dosage compensation process is not needed for the significance of the study.

The second stretch is the authors state that the effects of aneuploidy are fundamentally different than in other eukaryotes. However, the aneuploids analyzed are for small chromosomes because larger chromosomes are not found in an aneuploid state. This is the case even though the larger chromosomes are still smaller fractions of the whole genome than individual chromosomes in most other species. This suggests that larger aneuploids are indeed detrimental. The absence of their study leaves a void in terms of why that is the case. These claims should be removed from the manuscript because the study was not comprehensive enough to support this claim.

The issue is raised that mRNAs and protein levels do not necessarily correspond. This is widely known for steady state levels of most gene products with the exception of transcription factors and some other proteins (Ori et al., 2016, Genome Biol 17:47). Thus, the data presented do not provide evidence of any active process that accounts for the differences. The discussion of this topic should not be emphasized as particularly novel.

Reference 1 (Siegel and Amon) should be replaced with Birchler and Veitia, 2021, Cytogenetics and Genome Research 161: 529-550, which is more recent, more historically accurate, more comprehensive, and illustrates how the effects of aneuploidy intersect with other biological and physical chemistry principles.

Specific comments:

Lines 377 to 378: As noted previously, these references do not support the statements in this sentence. In the response, the authors say they substituted these for Sun et al, Hou et al., Raznahan et al, and Zhang et al, but there is no change. The references 46-49 should be deleted. Indeed, Hou et al showed that 47 and 49 needed re-interpretation. Reference 46 (Hwang et al) claims there are few trans effects but examination of their data shows they are abundant.

As noted in the previous review, single celled organisms such as yeast can select changes in absolute dosage to increase flux through certain pathways (Conant and Wolfe, 2007; Mol Syst Biol 3: 129). The case here of Leschmania may have some parallels to this situation, where there is polymorphism for aneuploid chromosomes that might provide a selective advantage in some circumstances. Nevertheless, under most circumstances, aneuploidy in yeast is detrimental and there is evidence for dosage sensitivity of the regulatory machinery that produces trans effects across the genome. Claims that there are few trans effects in yeast (mainly from the Amon group) are not supported by the fact that dosage sensitive genes were retained following the whole genome duplication in the evolutionary past of this species and are underrepresented in CNV of yeast (Hakes et al 2007, Genome Biology 8: R209) and studies of haploinsufficiency (Papp et al Nature, 2003) and overexpression (Robinson et al 2021). The dosage sensitive genes are typically members of multicomponent regulatory complexes (and some others). Thus, there are indeed dosage effects of aneuploids and selected genes in yeast. Now, it is potentially possible that the constitutive expression in Leischmania has evolved a different process. However, the fact that larger aneuploids do not appear to be so polymorphic as for the shorter chromosomes and were not studied suggests that the available data cannot support a claim that there is a difference. Further study would be needed but is beyond the scope of this submission.

The authors need to pull back from claims of some type of active dosage compensation and of a fundamental difference with other eukaryotes because the data do not support these claims or at least leave ambiguity. The documentation that there is polymorphism for several shorter chromosome aneuploidies that could potentially have adaptive roles is sufficiently important and interesting on its own.

Reviewer #3: (No Response)

**Part III – Minor Issues: Editorial and Data Presentation Modifications**

Reviewer #1: No major issues

Reviewer #2: See above.

Reviewer #3: (No Response)

PLOS authors have the option to publish the peer review history of their article (what does this mean?). If published, this will include your full peer review and any attached files.

Reviewer #1: No

Reviewer #2: No

Reviewer #3: No

Figure Files:

Data Requirements:

Reproducibility:

References:

---

## [Editor Report · Decision Letter 1]

30 Aug 2022

Dear Dr Cuypers,

We are pleased to inform you that your manuscript 'Four layer multi-omics reveals molecular responses to aneuploidy in Leishmania' has been provisionally accepted for publication in PLOS Pathogens.

Best regards,

Christian R. Engwerda

Section Editor

PLOS Pathogens

Christian Engwerda

Section Editor

PLOS Pathogens

Kasturi Haldar

Editor-in-Chief

PLOS Pathogens

orcid.org/0000-0001-5065-158X

Michael Malim

Editor-in-Chief

PLOS Pathogens

orcid.org/0000-0002-7699-2064
---

## [Editor Report · Acceptance letter]

20 Sep 2022

Dear Dr Cuypers,

We are delighted to inform you that your manuscript, "Four layer multi-omics reveals molecular responses to aneuploidy in <iLeishmania</i>," has been formally accepted for publication in PLOS Pathogens.

Best regards,

Kasturi Haldar

Editor-in-Chief

PLOS Pathogens

orcid.org/0000-0001-5065-158X

Michael Malim

Editor-in-Chief

PLOS Pathogens

orcid.org/0000-0002-7699-2064